# Activation of Steroidogenesis, Anti-Apoptotic Activity, and Proliferation in Porcine Granulosa Cells by *RUNX1* Is Negatively Regulated by H3K27me3 Transcriptional Repression

**DOI:** 10.3390/genes11050495

**Published:** 2020-04-30

**Authors:** Yuyi Zhong, Liying Li, Yingting He, Bo He, Zhonghui Li, Zhe Zhang, Hao Zhang, Xiaolong Yuan, Jiaqi Li

**Affiliations:** 1Guangdong Provincial Key Lab of Agro-Animal Genomics and Molecular Breeding, National Engineering Research Centre for Breeding Swine Industry, College of Animal Science, South China Agricultural University, Guangzhou 510642, China; yyzhong04@scau.edu.cn (Y.Z.); leeleeying@163.com (L.L.); heytada@sina.com (Y.H.); hbo@stu.scau.edu.cn (B.H.); zhezhang@scau.edu.cn (Z.Z.); zhanghao@scau.edu.cn (H.Z.); 2Institute of Animal Biotechnology, Xinjiang Academy of Animal Science, Urumqi, Xinjiang 830000, China; 13660659037@163.com

**Keywords:** H3K27me3, *RUNX1*, granulosa cells, steroidogenesis, cell apoptosis and proliferation, antral follicles, pigs

## Abstract

H3K27me3 is an epigenetic modification that results in the repression of gene transcription. The transcription factor RUNX1 (the runt-related transcription factor 1) influences granulosa cells’ growth and ovulation. This research uses ELISA, flow cytometry, EDU, ChIP-PCR, WB and qPCR to investigate steroidogenesis, cell apoptosis, and the proliferation effect of *RUNX1* in porcine granulosa cells (pGCs) as regulated by H3K27me3. Decreased H3K27me3 stimulates the expression of steroidogenesis-related genes, including *CYP11A1*, *PTGS2*, and *STAR*, as well as prostaglandin. H3K27me3 transcriptionally represses *RUNX1* here, whereas RUNX1 acts as an activator of *FSHR*, *CYP11A1*, and *CYP19A1*, promoting the production of androgen, estrogen, and prostaglandin, as well as increasing anti-apoptotic and cell proliferation activity, but decreasing progesterone. Both the complementary recovery of the H3K27me3 antagonist with the siRUNX1 signal, and the H3K27me3 agonist with the RUNX1 signal to maintain RUNX1 lead to the activation of *CYP19A1*, *ER1*, *HSD17β4*, and *STAR* here. Androgen and prostaglandin are significantly repressed but progesterone is markedly increased with the antagonist and siRUNX1. Prostaglandin is significantly promoted with the agonist and RUNX1. Furthermore, H3K27me3-RUNX1 affects the anti-apoptotic activity and stimulation of proliferation in pGCs. The present work verifies the transcriptional suppression of *RUNX1* by H3K27me3 during antral follicular development and maturation, which determines the levels of hormone synthesis and cell apoptosis and proliferation in the pGC microenvironment.

## 1. Introduction

The ovary is an important organ in the female reproductive system which is responsible for follicular growth, ovulation, and the synthesis of sexual hormones under the regulation of the hypothalamic–pituitary–gonadal (HPG) axis. Granulosa cells (GCs), as a component in follicular units, are critical for supporting ovarian function and determining follicular fate through the function of estrogens, growth factors, and cytokines [1]. These factors have been categorized as follicular survival factors, including estradiol estrogen receptor α/β [2], insulin-like growth factor 1 (IGF-1) [3], epidermal growth factor [4], and follicular atresia factors, such as FOXO3/ FOXO1 [5,6], inhibin [7], and TNF-associated receptors [8]. GCs are autocrine, paracrine, and endocrine units. The autocrine/paracrine factors have been proven to regulate follicular growth and development and in vitro steroidogenesis in cultured GCs, such as through IGF-1 [9] and endothelin-1 [10], which were significantly increased in preovulatory follicles in vivo. Growth hormones are classical paracrine and endocrine factors in GCs that regulate steroidogenesis, folliculogenesis, oocyte maturation, and the response to human menopausal gonadotropin (hMG) stimulation in the induction of ovulation [11]. Furthermore, steroidogenesis is another main function of GCs in response the stimulation of the FSH and LH, maintaining follicle growth and causing oocyte maturation and ovulation to proceed, alongside estrogen, androstenedione, testosterone and progesterone [12], and in addition to prostaglandin E2 (PGE)/F2α (PGF) [13]. Hence, steroidogenesis involves numerous complicated biological processes.

Along with HPG axis hormones, ovarian steroid hormones are important during the processes of follicular growth and maturation. The HPG axis hormones regulate follicular growth and development from nine weeks after birth [14]. Along with the high concentrations of FSH and LH during puberty, estrogen secretion peaks in the two days before puberty, and the normally low level of progesterone increases until the formation of the luteal corpus [15]. PGE and PGF significantly increases during follicular maturation [13] and are induced by LH to act on follicular development [16]. Besides FSH and LH, human chorionic gonadotropin (hCG) also acts on follicular growth and maturation by inducing progesterone and 17-hydroxyprogesterone [17]. Hormones of the HPG axis influence steroidogenesis via the regulation of steroidogenesis related key genes [18,19]. Holly (2017) summarized the relationship and synthesis pathway of steroidogenesis, showing that progesterone is the precursor of the intermediate products androstenedione and dehydroepiandrosterone, which are precursors of estrone and testosterone to be catalyzed into estradiol. Genes control the major molecular mediating steps, including *STAR*, *CYP11A1*, *HSD3β1*, *CYP17A1*, *CYP19A1*, *HSD17β1*, and *HSD17β7* [20]. 

Compared with the rate-limiting molecules directly regulating steroidogenesis in GCs, epigenetics signifies an indirect but necessary means of regulation. Histone post-translational modification is a form of epigenetic regulation that includes modification by methylation, acetylation, or ubiquitination [21]. In histone methylation, the lysine or arginine of the N terminal is methylated by histone methyltransferase. Histone lysine 4 trimethylation (H3K4me3) and histone lysine 27 trimethylation (H3K27me3) have been reported to act as activators and repressors of gene transcription [22]. It has been reported that increasing estrogen during rat puberty is the result of a loss of H3K27me3 and H3K9 modification, but of greater H3K4me3 modification of the *CYP11A1* promoter [23]. Therefore, histone methylation might play an important role in folliculogenesis, steroidogenesis, oocyte maturation, and ovulation via epigenetic regulation of gene transcription. 

The runt-related transcription factor 1 (RUNX1) is a nucleus transcription factor which has been reported to stimulate cell proliferation and progesterone secretion in goat GCs [24]. RUNX1 is necessary for ovulation in mice, and its expression is significantly upregulated via the miR-101 repression of histone methyltransferase EZH2, resulting in reduced H3K27me3 in the *RUNX1* promoter region [25]. These results suggest that H3K27me3 might regulate the transcription of RUNX1 in the biological function of GCs and the development of follicles. Consequently, the objective of this research is to investigate the role of *RUNX1* in regulating steroidogenesis, cell apoptosis and proliferation in porcine granulosa cells (pGCs) under the epigenetic regulation of H3K27me3. The results of this study offer further perspectives on the epigenetic regulation of histone methylation in GC growth, follicular development, and ovulation.

## 2. Materials and Methods

### 2.1. Ethics Statement

The animal experiments were conducted according to the Regulations for the Administration of Affairs Concerning Experimental Animals (Ministry of Science and Technology, China) and were approved by the Animal Care and Use Committee of South China Agricultural University, Guangzhou, China (approval number: 2018B116).

### 2.2. Porcine Granulosa Cell Culture 

The porcine ovaries used to culture GCs were collected from a slaughterhouse. Healthy ovaries were chosen and kept in saline on ice and sent to a sterile room as soon as possible before being washed with 37 °C physiological saline. Then, the follicular liquid in 3–5 mm follicles was absorbed with a 1 mL sterile syringe needle. The follicular liquid mixture (containing follicular liquid and cumulus-oocyte complexes) was stored with Dulbecco’s modified eagle medium (DMEM) (Hyclone, Logan, UT, USA) in a 15-mL sterile centrifuge tube. A total of 2 mL of the follicular liquid in each tube was centrifuged at 1000 rpm for 10 min to collect the cells, which were then washed with DMEM twice. Finally, the cells in each tube were cultured in a 75-mm^2^ cell culture flask with 15 mL of the DMEM complete medium supplemented with 10% fetal bovine serum (Hyclone, Logan, UT, USA) and 1% penicillin and streptomycin (Thermo Fisher, Waltham, CF, USA). The medium was refreshed after two days to harvest the pure porcine granulosa cells (pGCs) because pGCs adhere and cover the flask but oocytes are suspended in the culture medium. When the pGCs reached full growth in the flask, they were able to be used in further processes. The methods above have been referenced from Xin et al. (2019) [26].

### 2.3. Expression Profiles of H3K27me3 in Different Stage Follicles 

Healthy ovaries (*n* = 4) from two female pigs were collected from a slaughterhouse, cleaned, and stored on ice. Follicles that had a bright and smooth follicular membrane, abundant vasculature, and were filled with clear follicular liquid were chosen and isolated from the ovaries for WB. According to the follicular size, the follicles were divided into medium-sized, big-sized, and mature follicles [27]. The follicles in each group were from more than three replications. The protein of the follicles was extracted using a protein extraction kit (P003, Beyotime Biotechnology, Shanghai, China). The protein concentration was measured via a BCA kit (P0012, Beyotime Biotechnology, Shanghai, China). Then, the H3K27me3 and RUNX1 production of each sample were quantified by WB [28]. Equal amounts of proteins in each group were isolated by SDS-PAGE and electroblotted on to polyvinylidene difluoride membranes. The reaction was blocked-up by phosphate buffer saline (PBS) with 5% fat-free milk and 1% Tween-20 for 60 min. The membranes were then incubated and reacted with the primary antibodies overnight at 4 ℃. The dilutions of anti-H3K27me3 (#07-449, Millipore, Germany) and anti-RUNX1 (ab92336, Abcam, Cambridge, UK) were 1:5000. The anti-GAPDH (1:10,000, Sigma, St. Louis, MO, USA) was employed as an internal control. The secondary antibodies were incubated for 60 min at room temperature. The bonding of antibody-specific protein was visualized with an ECL-PLUS Kit (Amersham Biosciences, Piscataway, NJ, USA). The gray scale values of the bands were calculated using the ImageJ software package. The H3K27me3 and RUNX1 relative gray values in the qWB were calculated as the ratio of the H3K27me3 gray value/ GAPDH gray value and RUNX1 gray value/ GAPDH gray value.

### 2.4. H3K27me3 Inhibitor and Activator 

In order to interfere with the production of H3K27me3 in pGCs, GSK-126 (#1346574-57-9, MedChemExpress, Monmouth Junction, NJ, USA) and GSK-J4 (#1373423-53-0, MedChemExpress, Monmouth Junction, NJ, USA) were chosen to be the antagonist and agonist of H3K27me3, respectively [29]. The doses of antagonist GSK-126 and agonist GSK-J4 were 6 and 2 nM. The negative controls of antagonist GSK-126 and agonist GSK-J4 were 6 and 2 nM of a dimethyl sulfoxide (DMSO) solvent that was used to dissolve the antagonist and agonist. The effect of the drugs was measured by WB. The performances of WB were the same as above. The antibodies of H3K27me3 were purchased from Millipore (#07-449, Millipore, Germany). GAPDH (Sigma, St. Louis, MO, USA) was used as reference protein. Each group featured three replications for cell treatments and two replications for WB.

### 2.5. Reconstructed Vectors and RNAi Fragments of RUNX1 

According to the reference sequence of *RUNX1* (XM 021068413.1), the coding sequences (CDS) were amplified and used to construct the vectors for overexpression. The primers used to amplify RUNX1 are presented in Appendix A. The RUNX1 CDS fragment was ligated into the pcDNA3.1 (+) vector to obtain pcDNA3.1-RUNX1. The siRNAs of RUNX1 were synthesized by the BioRio company (Guangzhou, China). The RNAi sequences are shown in Appendix A. The overexpression vectors and RNAi sequences were transfected into pGCs in a 6-well cell culture plate. The amounts of the recombinant plasmids used were 500, 750, 1000, and 1500 ng. The concentrations of the siRUNX1 fragments were 50, 75, 100, and 150 nM. The transfection groups of the pcDNA3.1 plasmids and siRNA negative sequences (siNC) were the controls of pcDNA3.1-RUNX1 and siRUNX1. Each group featured three replications. Lastly, qPCR was used to measure gene overexpression and interference. The total RNA was reverse transcribed using a PrimeScript RT Master Mix Synthesis Kit (Takara, Japan). The gene expression was quantitated by the measuring dye included in the Maxima SYBR Green qPCR Master Mix (2×) (Thermo Fisher, Waltham, CF, USA) for qPCR, and the relative expression was calculated using the 2^−ΔΔct^ method. 

### 2.6. Bioinformation Prediction and Chromatin Immunoprecipitation (ChIP)

The target gene *RUNX1* was found using the UCSC Genome Browser (https://genome.ucsc.edu) for pigs in February 2017 (Sscrofa11.1/susScr11). Then, using the human genome sequence (December 2013 (GRCh38/hg38)) and Chain and Net Alignments, the homologous human sequence for the *RUNX1* promoter and open reading frame was located. The homologous promoter sequence was obtained via “Chain Alignments”. Both of the H3K4me3 marks (often found near promoters) were found for seven cell lines from the ENCODE signal, and DNase I hypersensitivity was found in 95 cell types from the ENCODE signal in the ENCODE dataset of the human gene database. The H3K27me3 modification peak in swine RUNX1 DNA was obtained after verification of the predicted binding region in GRChistone 8/he38. Finally, primers (Appendix A) were designed and used in the ChIP-PCR assay.

The relationship between H3K27me3 and RUNX1 was identified via ChIP. The H3K27me3 antibodies were purchased from Millipore (#07-449, Millipore, Germany). The operations can be referred to as in [30] and the protocol included with the Pierce ChIP kit (cat. 26156, Thermo Fisher, Waltham, CF, USA): (1) more than 2 × 10^7^ pGCs were crosslinked by 1% terminal concentration paraformaldehyde (cat. C104188, Aladdin). (2) The cells were lysed and digested with Lysis Buffer 1, MNase Digestion Buffer, and micrococcal nuclease. (3) The ChIP reaction of the supernatant and antibody was carried out through overnight incubation. For this, 10 μL of the anti-RNA polymerase II antibody was added in the positive control IP, while the negative control IP included the addition of 2 μL of rabbit IgG, and 10 μg of the antibody was added in the target-specific IP. (4) According to the elution and DNA recovery, the purification products of the Input DNA, IP DNA, positive DNA, and negative DNA were collected. (5) For qPCR, the primers used to amplify the binding region of the predicted target genes are listed in Appendix A. qPCR was carried out with a in a CFX ConnectTM Fluorescence Quantitative PCR Instrument (cat. 1855200, BIORAD). Each group featured three repeats. (6) The relative expression of each gene was calculated using the 2^−ΔΔct^ method. The percentage of the input recovery value is equal to the power (2 × −(target IP value − 100% input value)) × 100, and the IgG value is equal to power (2 × −(target IgG value − 100% input value)) × 100. 

### 2.7. qPCR

The genes, including *FSHR* (NM 214386.3), *LHR* (JN 120794.1), *ER1* (NM 214220.1), *ER2* (NM 001001533.1), *CYP11A1* (NM 214427.1), *CYP19A1* (NM_214429), *HSD17β1* (EU 153250.1), *HSD17β4* (NM_214306.1), *PTGS2* (NM 214321.1), and *STAR* (NM 213755.2), were chosen to investigate the effect of H3K27me3, RUNX1, and the coupled effects of H3K27me3-RUNX1 in steroidogenesis. The primers of each gene are listed in Appendix A. The pGCs were assayed after 48 h for treatment of pcDNA3.1-RUNX1, siRUNX1, antagonist GSK-126 with pcDNA3.1-RUNX1, antagonist GSK-126 with siRUNX1, agonist GSK-J4 with pcDNA3.1-RUNX1, and agonist GSK-J4 with siRUNX1. The control groups were pcDNA 3.1, siNC, antagonist GSK-126 with pcDNA3.1, antagonist GSK-126 with siNC, agonist GSK-J4 with pcDNA3.1, and agonist GSK-J4 with siNC. Each group featured three repeats. The total RNA of each group was extracted with the traditional method using the TRIzol regent (TaKaRa, Tokyo, Japan). The quality of total RNA was assessed using agarose gel and a nuclei acid spectrometer (NanoDrop One, Thermo Fisher, Waltham, CF, USA). The RNA was reverse transcribed into cDNA and used for qPCR. GAPDH was used as a reference gene, and the relative expression of the detected genes were calculated using the 2^−ΔΔct^ method. The qPCR of each group featured three replications. 

### 2.8. Western Blotting 

The protein levels of genes that were indicated by qPCR to be significantly upregulated or downregulated by RUNX1, and H3K27me3-RUNX1 were also analyzed in pGCs. This assay included the inhibition of RUNX1 with siRUNX1 in pGCs which were exposed to hCG, antagonist GSK-126, and agonist GSK-J4 after 6 h of transfection. Antibodies against RUNX1 (Abcam), PTGS2 (bs-0732R, BioSS ANTIBODIES, Woburn, USA), CYP11A1 (bs-10099R, BioSS ANTIBODIES, Woburn, MA, USA), CYP19A1 (bs-1292R, BioSS ANTIBODIES, Woburn, USA), HSD17β1 (bs-3855R, BioSS ANTIBODIES, Woburn, MA, USA), HSD17β4 (bs-11296R, BioSS ANTIBODIES, Woburn, USA), ER1 (bs-2098R, BioSS ANTIBODIES, Woburn, USA), and ER2 (bs-0116R, BioSS ANTIBODIES, Woburn, USA) were used to measure the corresponding protein levels by WB. The dilutions of the primary-antibodies were 1:1000, and the dilutions of the secondary-antibodies were 1:2500. The negative control was transfected with siNC and simultaneously treated with hCG, antagonist GSK-126, and agonist GSK-J4. The WB of each group featured two replications. The performances of the WB were the same as above. The protein relative gray value in group of siRUNX1/hCG (−) and siRUNX1/ hCG (+) was calculated as the ratio of protein gray value / GADPH gray value. In the situations of treatments of antagonist GSK-126 and agonist GSK-J4, the protein relative gray value was equal to the ratio of protein relative gray value of siRUNX1/ protein relative gray value of siNC. The up-regulated protein should be a qualified protein relative gray value of antagonist GSK-126 or agonist GSK-J4 > 1 and greater than protein relative gray value of antagonist NC or agonist NC. The down-regulated was protein relative gray value of antagonist GSK-126 or agonist GSK-J4 < 1 and less than protein relative gray value of antagonist NC or agonist NC.

### 2.9. Hormone Detection 

The concentrations of androgen, estrogen, progesterone, and prostaglandin in the cell cultural media were measured with ELISA [31]. The medium of each well was collected after 48 h of drug treatment and transfection with the plasmids and/or RNAi fragments. The media samples were centrifuged at 3000 rpm for 20 min. The supernatant was directly used for hormone analysis or stored at −80 °C. The Porcine Androgen ELISA kit (JL26487), Porcine Estrogen ELISA kit (Java 10508), Porcine Progesterone ELISA kit (JL36738), and Porcine Prostaglandin ELISA kit (JL21995) were purchased from JiangLai Biotech (Shanghai, China). The methods were carried out according to the manufacturer’s protocols. Then, 50 μL of standard samples or tested samples was added to each well of the plate, but with no samples in the blank wells. The wells were covered with a diaphragm seal and placed in water baths at 37 °C for 30 min. The discarded liquid and washed the samples with 350 μL scrubbing solution five times. Then, 50 μL of the biomarker antibody was added to each tested sample but not added to this to the standard samples or blank wells. They were then covered with a diaphragm seal and placed in water baths at 37 °C for 30 min. The liquid was discarded, and the samples were washed with a scrubbing solution five times. Next, 100 μL of the horseradish peroxidase labeling detection antibody was placed into the wells with the standard and tested samples, where the liquid was then thrown out and the samples were left to stand still for 1 min with 350 μL of the scrubbing solution. This was repeated five times. Next, 50 μL of substrate was added to each well, followed by incubation at 37 °C in the dark for 15 min. Lastly, 50 μL of the stop buffer was added and the OD value was read at a 450 nm after 15 min. The standard linear curve was drawn according to the standard sample values and used this was used to calculate the concentration of each sample. The hormone detection of each group featured four repeats.

### 2.10. Cell Apoptosis and Proliferation 

The cells used for hormone detection were collected to analyze apoptosis using annexin V/PI. The cells were detected in two sections. The first section was the cells transfected with RUNX1-pcDNA3.1 plasmid, pcDNA3.1 plasmid, siRUNX1 sRNA, and control siNC. The second section was the four groups from the first section, along with the treatment of antagonist GSK-126 or agonist GSK-J4. The cells were collected with tryptase (Gibico, Grand Island, NY, USA) 48 h after transfection and washed three times with PBS (Hyclone, Logan, UT, USA). The cells were then stained with Annexin V/PI and incubated in the dark for 15–30 min. The proportion of apoptotic cells was detected using flow cytometry (FCM) after 1 h. The cell apoptosis detection of each group featured three repeats. The proportion of cell apoptosis in each group was visualized according to the output from the FCM scatter plots. The proportion of apoptotic cells was equal to the sum of the upper and lower right quadrants.

The 5′-ethynyl-2′-deoxyuridine (EDU) was used to characterize pGCs cell proliferation. The experiment followed the manufacturer’s protocol for the Cell-Light EdU Apollo 488 In Vitro kit (C10310-3, RIBOBIO, Guangzhou, China). EDU detection was carried out 24 h after transfection. Briefly, pGCs in a 48-well plate were incubated with the EDU medium for 2 h (DMEM/EDU = 1000:1). They were washed with PBS for twice and fixed with 80% acetone for 30 min. They were then washed with PBS for a further three times and permeabilized with 0.5% Triton X-100 for 30 min. They were washed with PBS for three times again and incubated with 1× Apollo for 30 min in the dark. The cells were then washed with PBS for at least five times in a 200-rpm shaker, for 5 min each time. Then, 1× DAPI was added and calculated the total cells and proliferated cells in the four different horizons of each well as observed under an inverted fluorescence microscope (TE2000-U inverted microscope, Nikon Instruments, Tokyo, Japan). Each treatment featured three replicates. Finally, the cell proliferation rates were calculated as the ratio of the sum of proliferated cells to the sum total cells.

### 2.11. Data Analysis 

The data are expressed as the mean ± SEM. Statistical comparisons between the different groups were performed using a one-way ANOVA. Paired data were evaluated by Student’s *t*-test with the GraphPad Prism 7.0 software (San Diego, CA, USA). For this, * indicates *p* < 0.05 and ** indicates *p* < 0.01. 

## 3. Result

### 3.1. Function of H3K27me3 in Steroidogenesis of pGCs

To explore the biological functions of H3K27me3 in the development of follicles, the expression of H3K27me3 was first detected in medium-sized follicles (~5 mm), big-sized follicles (~8 mm), and graafian follicles (~13 mm) (Figure 1). The diameters of these follicles change significantly in length (Figure 1a), and the protein counts of H3K27me3 were highest in the medium-sized follicles but lowest in the graafian follicles (Figure 1b,c). Moreover, the protein counts of H3K27me3 in the graafian follicles were markedly lower than in the medium-sized follicles and big-sized follicles (Figure 1c). These results suggest that H3K27me3 might play a critical role during the development of follicles. The protein counts of RUNX1 showed that H3K27me3 and RUNX1 is do not feature an absolutely inverse correlation, but RUNX1 is the highest expressed in graafian follicles that H3K27me3 is the lowest expressed (Figure 1b,c).

To further investigate the regulation of the development of follicles by H3K27me3, the expression of H3K27me3 was upregulated and downregulated in GCs with the inhibitor antagonist GSK-126 and agonist GSK-J4 for H3K27me3. The results of the WB showed 6 nM of the antagonist and 2 nM of the agonist significantly and respectively repressed and promoted the yields of H3K27me3 (Figure 2a). Compared to the pcDNA3.1 control, all of the 500-, 750-, 1000- and 1500-ng pcDNA3.1-RUNX1 plasmids significantly increased *RUNX1* at the mRNA level (Figure 2b), and the results of WB further verified that the 1000-ng plasmids increased RUNX1 at the protein level (Figure 2c). Compared to siNC control, all of the 50- (*p* > 0.05), 75- (*p* > 0.05), 100- (*p* < 0.05) and 150-nM (*p* > 0.05) siRUNX1 fragments decreased *RUNX1* at the mRNA level (Figure 2d), and results of WB further verified that 100 nM reduced RUNX1 on a protein level (Figure 2e). Consequently, these conditions were used in further experiments, i.e., 6 nM of the antagonist GSK-126 and antagonist-NC, 2 nM of the agonist GSK-J4 and agonist-NC, 1000 ng of the pcDNA3.1-RUNX1 recombined plasmids and pcDNA3.1 plasmids, and 100 nM of the siRUNX1 and siNC fragments.

Compared to the antagonist control, it was found that the downregulated H3K27me3 significantly increased the mRNA levels of *CYP11A1, ER1*, *ER2*, *FSHR*, *HSD17β1*, *PTGS2, STAR,* and *RUNX1*(*p* < 0.01), and *HSD17β4* (*p* < 0.05), and significantly decreased the mRNA levels of *LHR* (*p* < 0.01). Compared to the agonist control, the upregulated H3K27me3 significantly increased the mRNA levels of *ER2* (*p* < 0.01), *FSHR* (*p* < 0.05), and *LHR* (*p* < 0.01)*,* and significantly decreased the mRNA levels of *PTGS2*, and *RUNX1*(*p* < 0.01) (Figure 3a). Compared to antagonist-NC, many genes were increased with antagonist, but it was found that only prostaglandin (*p* < 0.01) significantly improved with antagonist, which was significantly decreased with the agonist (*p* < 0.01) (Figure 3b). 

### 3.2. Function of H3K27me3 Target RUNX1for Steroidogenesis in pGCs

The location of the potential binding site of H3K27me3 on the RUNX1 promoter was predicted at chr13: 198,653,501–198,653,648 bp (Appendix A). The comparison of the expression of the positive control of anti-RNA polymerase II and negative control of IgG is a popular method for verifying the combination between specific proteins and the target DNA. The positive control in this experiment showed the expression of GAPDH was higher than IgG (Figure 4). As shown in Figure 4a,b, the promoter occupying RUNX1 with the input and IP template was more than that of the negative control. H3K27me3 acts as a suppressor to regulate RUNX1 transcription (Figure 4c). Therefore, the molecular regulation characteristic is settled in that H3K27me3, which, as a component of the nucleosome, represses RUNX1, thereby regulating downstream targets to influence pGC function.

The qPCR showed that, compared with the pcDNA 3.1 plasmids, *CYP11A1, CYP19A1, ER1, LHR, HSD17β4, PTGS2* and *RUNX1* (*p* < 0.01), and *FSHR* (*p* < 0.05)*,* at the mRNA level were significantly promoted, but *HSD17β1* (*p* < 0.01) significantly declined when RUNX1 was overexpressed. Compared to siNC, *ER1, HSD17β4, PTGS*, and *RUNX1* (*p* < 0.01), *CYP19A1, ER2, FSHR*, and *LHR* (*p* < 0.05) were markedly decreased (Figure 5a). Compared to siNC/ hCG (-), the proteins of ER2 (*p* < 0.01), HSD17β4 (*p* < 0.01), and RUNX1 (*p* < 0.05) were significantly repressed, while CYP19A1 (*p* < 0.01) was significantly increased with siUNX1. Compared to siNC/ hCG(+), the ER1 (*p* < 0.01), ER2 (*p* < 0.01), HSD17β1 (*p* < 0.05), and RUNX1 (*p* < 0.01) proteins were significantly activated (Figure 5b). The activation of key steroidogenesis genes at the protein level by RUNX1 leads to the promotion of androgen and estrogen, but also degradation of progesterone. Additionally, the secretion of prostaglandin was also increased (Figure 5c). Furthermore, it was discovered that higher *RUNX1* exhibited anti-apoptotic and stimulated cell proliferation properties, yet the inhibition of *RUNX1* stimulated apoptosis in pGCs (Figure 5d,e).

### 3.3. Effect of H3K27me3-RUNX1 Signals on Steroidogenesis in pGCs

The mRNA expression of key steroidogenesis-related genes was investigated. (1) With the treatment of antagonist GSK-126 together with *RUNX1* overexpression, *RUNX1, CYP11A1*, and *PTGS2* were significantly increased, but *CYP19A1, ER1, HSD17β1*, *HSD17β4*, and *STAR* were significantly decreased (Figure 6a). (2) When cells were treated with antagonist GSK-126 and siRUNX1 fragments, the mRNA levels of *CYP11A1*, *CYP19A1*, *ER1*, *ER2*, *HSD17β1*, *HSD17β4*, and *STAR* were significantly promoted, but *RUNX1* and *PTGS2* significantly declined (Figure 6a). It was found that H3K27me3 antagonist RUNX1 accumulation stimulated *RUNX1* and *PTGS2*, but repressed *CYP19A1*, *ER1*, *HSD17β1*, *HSD17β4*, and *STAR*, which were complementary recovered with siRUNX1. (3) When treated with agonist GSK-J4 and pcDNA3.1-RUNX1, *RUNX1*, *ER1*, *LHR*, *HSD17β4, STAR*, and *PTGS2* were significantly improved, but *ER2* and *CYP11A1* were markedly reduced. (4) When treated with agonist GSK-J4 together with siRUNX1, *FSHR*, and *CYP11A1* were obviously improved but *RUNX1*, *CYP19A1*, *ER1*, *HSD17β4*, and *PTGS2* were significantly decreased (Figure 6a). Consequently, the H3K27me3 agonist RUNX1 signal was available to promote *RUNX1*, *ER1*, *HSD17β4*, and *PTGS2* but to inhibit *CYP11A1,* which was reverted with siRUNX1 treatment. At the protein level, using antagonist GSK-126 and siRUNX1 significantly increased ER1, HSD17β4, CYP11A1, and RUNX1, but markedly inhibited CYP19A1. The results showed that antagonist GSK-126 blocked the inhibition of siRUNX1 to promote ER1, HSD17β4, CYP11A1, and RUNX1. Using agonist GSK-J4 and siRUNX1 significantly improved ER1, HSD17β1, and CYP11A1, but obviously reduced RUNX1 (Figure 6b). The couple’s inhibition of GSK-J4 and siRUNX1 led to an opposite activated effect on ER1, HSD17β1, and CYP11A1.

Treatment with antagonist GSK-126 and pCDNA3.1-RUNX1 promoted estrogen (*p* < 0.05), androgen (*p* > 0.05), progesterone (*p* > 0.05), and prostaglandin (*p* > 0.05). An antagonist GSK-126 with siRUNX1 significantly increased progesterone but significantly decreased androgen and prostaglandin (Figure 6c). The treatment with agonist GSK-J4 combined with pCDNA3.1-RUNX1 markedly increased prostaglandin, and agonist GSK-J4 together with siRNX1 obviously increased progesterone but repressed prostaglandin, both significantly and respectively. (Figure 6c). For the pGC phenotypes, along with the treatment of the H3K27me3 antagonist GSK-126 to release the inhibition of H3K27me3 on *RUNX1*, both overexpressed and inhibited *RUNX1*, resulting in repressed cell apoptosis in pGCs. When treated with agonist GSK-J4 to intensify the repression of H3K27e3 on *RUNX1*, cell apoptosis was decreased with *RUNX1* overexpression, however, cell apoptosis was stimulated when *RUNX*1 declined (Figure 6d). In other words, the stimulation of apoptosis by siRUX1 was reversed by the reduction of H3K27me3 with antagonist GSK-126, and the induction of cell apoptosis recovered with the accumulation of agonist GSK-J4 and siRUNX1. Along with an antagonist or agonist, overexpressed *RUNX1* significantly improved pGC proliferation and *RUNX*1 interference tended to inhibit cell proliferation. In brief, RUNX1 was a cell proliferation activator, but the cell proliferation effect of antagonist GSK-126 with *RUNX1* was better than that of for GSK-J4 with *RUNX1* (Figure 6e).

## 4. Discussion 

In this study, it was observed that H3K27me3 was significantly decreased during antral follicular growth and maturation (Figure 1). It was known to us that H3K27me3 together with H3K4me3, controls the compression of the inactive promoter and loosens the active promoter to switch off gene transcription [32]. As a result, H3K27me3 and H3K4me3 play essential roles in regulating organ development and growth [33,34], immunity cell function [35], and female puberty [36]. Rablo et al. (2008) pointed out that H3K27me3 is high in immature bovine oocytes and decreases after fertilization to the eight-cell stage, meaning that H3K27me3 is necessary for ovulation and preimplantation [37]. In this study, the expression of H3K27me3 was significantly decreased with the development of medium-sized follicles, big-sized follicles, and graafian follicles (Figure 1). Consequently, it was speculated that H3K27me3 may play an important role in the function of follicular growth, oocyte maturation, steroidogenesis, luteal corpus formation, maintenance, and lute lytic activity.

In this research, it was verified that H3K27me3 repressed prostaglandin via down-regulating *PTGS2* expression. *PTGS2* was upregulated when H3K27me3 declined and downregulated when H3K27me3 was activated (Figure 3a). Prostaglandin E2 (PGE2) is the product of prostaglandin-endoperoxide synthase 2 (PTGS2) and regulates ovulation in granulosa cells [38], and steroidogenesis [39] and luteal corpus formation and maintenance in granulosa corpus cells [40]. As a result, prostaglandin was significantly influenced by H3K27me3 (Figure 3b). 

ChIP-PCR demonstrated H3K27 trimethylation as a transcriptional repressor bound to the promoter region of *RUNX1*, thereby restricting transcription (Figure 4c). *RUNX1* is a nucleus transcription factor and is induced by LH in preovulatory follicles in mice and promotes progesterone production and the expression of *Cyp11a1* [41]. Decreased *RUNX1* also results in a decreased expression of *STAR*, *CYP11A1,* and *HSD3β* in goat GCs [24]. It was pointed out that *RUNX1* has a role as an activator in upregulating the expression of *CYP11A1*, *CYP19A1*, *ER1*, *FSHR*, *LHR*, *HSD17β4*, *PTGS2,* and *STAR* (Figure 5a). Furthermore, the translation into ER1, ER2, HSD171, PTGS2, and CYP11A1 is restricted when *RUNX1* is decreased by siRNA fragments (Figure 5b). Moreover, the function of *RUNX1* is in acting as a mediator to extend the response of steroidogenesis-related genes to human chorionic gonadotropin (hCG). The function of hCG is to stimulate ovulation via promoting nuclear maturation and follicular rupture [42]. Consequently, it has been speculated that increasing *RUNX1* would promote follicular maturation via the transcription factor RUNX1 and downstream target networks, such as RUNX1-mediated TGF-β in PI3K pathway [43] and RUNX1-mediated NOTCH4 signaling [44].

A previous report has demonstrated that STARD1, CYP11A1, HSD3B1, CYP17A1, CYP19A1, HSD17B1, and HSDB7 are the real limiting factors regulating hormone synthesis in pGCs [20]. It has been verified that *RUNX1* over-expression increased androgen, estrogen, and prostaglandin, but reduced progesterone (Figure 5c). The high production of androgen is associated with dominant follicular development and the maintenance of oocytes via a direct response to FSH-stimulated granulosa cells [45]. Additionally, estrogen exerts positive feedback on LH/FSH induction, androgen, and E2 secretion and promotes mitotic and anti-apoptotic effects on follicular cells [46]. The prostaglandin involved in the ovulation of follicles via PKB and MAPK3/1 induces mild cumulus expansion and meiosis resumption [47]. Thus, *RUNX1* would promote follicular development and ovulation via extending androgen, estrogen, and prostaglandin function in antral follicles. The interactions among the hormones has been investigated. McNatty et al. (1975) reported that prostaglandin F2α inhibits progesterone synthesis, but prostaglandin E2 was observed to stimulate progesterone levels in human GCs [48]. However, Fowkes et al. (2001) pointed out that progesterone and prostaglandin products are independent from each other [49]. Estrogen stimulates progesterone in pGCs, but androgen represses the hormone [50]. Androgen, estrogen, prostaglandin, and progesterone are all necessary for follicle development and maturation, the production of which is determined by cell states and follicular requirements. A decrease in progesterone may be due to the fact that it is a precursor of androgen and estrogen.

The use of an H3K27me3 antagonist and agonist to respectively decrease and increase H3K27me3 production in pGCs leads to more complicated results in terms of steroidogenesis gene expression and translation (Figure 6a,b). The intensive upregulation of *RUNX1* by decreasing H3K27me3 and the overexpression of RUNX1 did not further emphasize the activation of RUNX1, such that *LHR*, *ER1*, *CYP19A1*, *HSD17β1*, *HSD17β4*, and *STAR* were downregulated. The dual suppression of agonist GSK-J4 and siRUNX1 induced more gene inhibition, including in *RUNX1, ER1, CYP19A1, HSDβ4,* and *PTGS2*. However, the results of the WB showed that treatment with agonist GSK-J4 and siRUNX1 was likely to improve ER1, ER2, HSD17β1, CYP11A1, and CYP19A1, but reduced PTGS2 and RUNX1. Antagonist GSK-126 and siRUNX1 were able to promote ER1, HSD17β1, HSD17β4, CYP11A1, and RUNX1. Many reports have presented that excessive overexpression or exhaustive inhibition leads to negative feedback [51] and a compensatory effect [52] to support survival and normal metabolism, or else resulting in abnormal function [53], diseases [54], or even death [55]. Although the interaction between H3K27me3 and RUNX1 makes the gene expression and protein translation analysis less clear, the regulation patterns of androgen, estrogen, and prostaglandin were the same as when RUNX1 was upregulated and downregulated, for both cases of antagonist GSK-126 or agonist GSK-J4 treatment (Figure 6c). That is to say that the syntheses of androgen, estrogen, and prostaglandin were dependent on changes in RUNX1, but not necessarily H3K27me3.

The molecular mechanisms of the H3K27me3-RUNX1 signal that participate in follicular growth and maturation may be that H3K27me3 inhibits the transcription of RUNX1 to regulate the expression of the steroidogenesis-related gene and suppresses the synthesis of estrogen, androgen, and prostaglandin in the pGCs. The higher level of hormones in overexpressed *RUNX1* groups benefit pGC proliferation and increased anti-apoptotic activity (Figure 5e and Figure 6e). Previous reports have pointed out that proliferation capability is improved in GCs when they are exposed to E2, which is associated with the exposure of *Cxs*, *LHR*, *FSHR*, and *CYP19A1* [56]. Additionally, the blocking of androgen–androgen receptor signaling induces granulosa cell apoptosis and leads to reduced progesterone levels [57]. PGE2 is a key factor related to follicular ovulation, in that PGE2 induces expression of cumulus expansion-related genes (*HAS2* and *TNFAIP6*), the steroidogenesis-related gene *CYP11A1*, and the prostaglandin production gene *PTGS2* [58]. Consequently, the findings of this research emphasize the essential role of H3K27me3 RUNX1 signaling on antral follicular granulosa cell growth, and on the processes of prominent follicular selection, development, and maturation.

## 5. Conclusions

In conclusion, *RUNX1* dynamically increases during antral follicular growth and maturation, resulting from the gradually released transcriptional repression of H3K27me3. The increase of transcription factor RUNX1 improves estrogen, androgen, and prostaglandin secretion through regulating the expression of genes related to steroidogenesis, such as *FSHR, LHR, CYP11A1,* and *STAR*. High levels of hormones have an anti-apoptotic effect and stimulate cell proliferation in pGCs, which might participate in prior antral follicular growth and maturation. These findings broaden our understanding of the influence of the post-translational epigenetic regulation of histones in the functioning of granulosa cell function, and also further elucidating the regulatory mechanism of the transcription factor RUNX1 on hormone synthesis, cell apoptosis, and the proliferation in granulosa cells during antral follicular growth and maturation.

## Figures and Tables

**Figure 1 genes-11-00495-f001:**
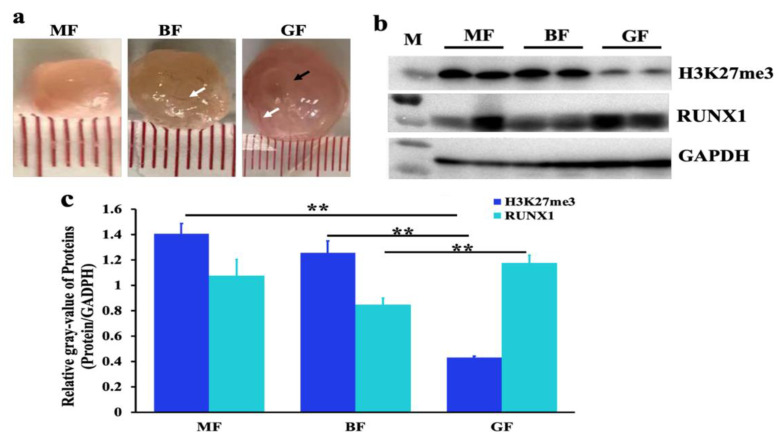
H3K27me3 and RUNX1 in different stages of follicles. (**a**) Follicles used for Western blotting (WB). MF, medium-sized follicle; BF, big-sized follicle; GF, graafian follicle. The white arrow indicates the blood vessels. The black arrow indicates the obvious ovulation hole in the mature follicle. (**b**,**c**) The relative gray values of H3K27me3 and RUNX1 in medium-sized follicles, big-sized follicles, and graafian follicles, normalized using the H3K27me3, RUNX2, and GAPDH gray values, respectively. The ** indicates *p* < 0.01.

**Figure 2 genes-11-00495-f002:**
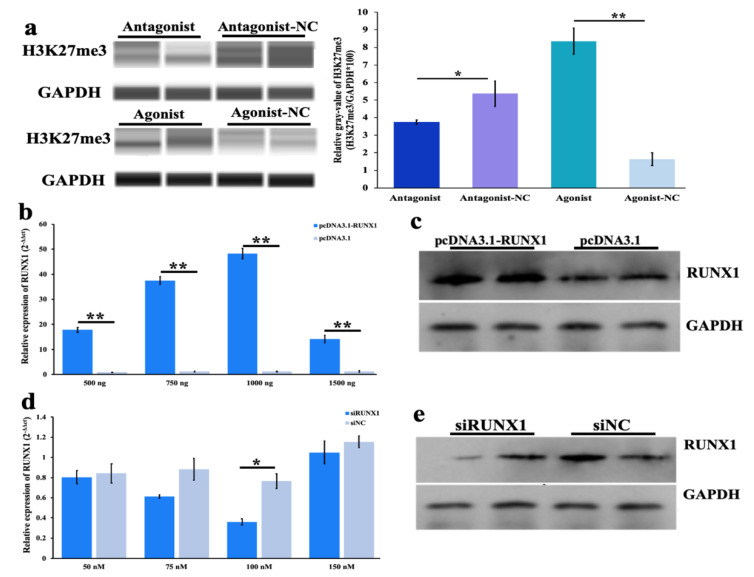
The activation and interference of H3K27me3 and the runt-related transcription factor (RUNX1) in porcine granulosa cells (pGCs). (**a**) The product of H3K27me3 with 6 nM of the antagonist GSK-126 and antagonist-NC, 2 nM of the agonist GSK-J4 and agonist-NC, respectively; H3K27me3 relative gray value = H3K27me3 gray value/ GAPDH gray value. The * indicates *p* < 0.05 and ** indicates *p* < 0.01. (**b**) The mRNA level of *RUNX1* when transfected with 500-, 750-, 1000- and 1500-ng pcDNA3.1-RUNX1 and control of pCDNA3.1 plasmid; (**c**) The protein level of RUNX1 when transfected with 1000 ng of the pcDNA3.1-RUNX1 and pCDNA3.1 plasmids. (**d**) The mRNA level of *RUNX1* when transfected with 50, 75, 100 and 150 nM of the siRUNX1 and control of siNC fragments. (**e**) The protein level of RUNX1 when transfected with 100 nM of the siRUNX1 and siNC fragments.

**Figure 3 genes-11-00495-f003:**
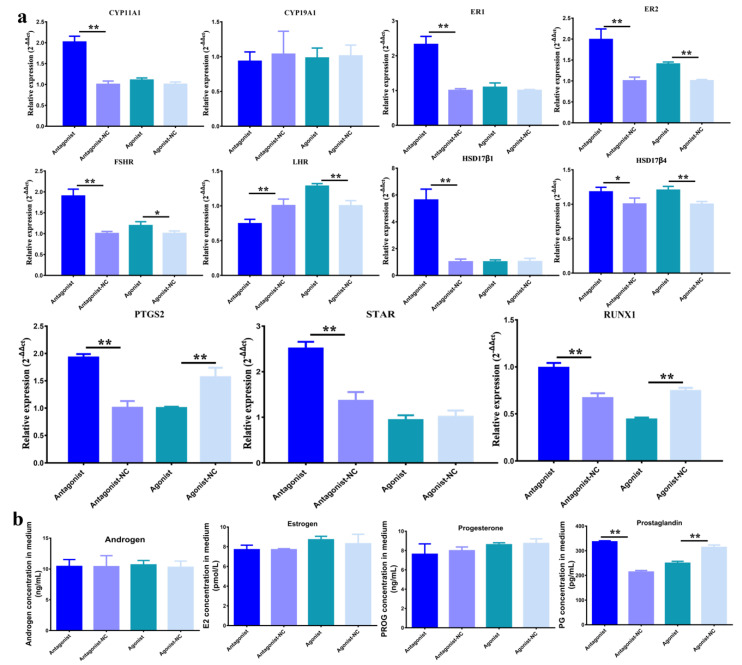
The impact of H3K27me3 on steroidogenesis in pGCs. (**a**) The expression patterns of steroidogenesis related genes when treated with agonist GSK-126 or agonist GSK-J4 in pGCs. Antagonist-NC and agonist-NC were the controls of antagonist GSK-126, and agonist GSK-J4, respectively. The * indicates *p* < 0.05 and ** indicates *p* < 0.01. (**b**) The synthesis of hormones androgen, estrogen, prostaglandin, and progesterone with antagonist GSK-126, antagonist-NC, agonist GSK-J4, and agonist-NC in pGCs.

**Figure 4 genes-11-00495-f004:**
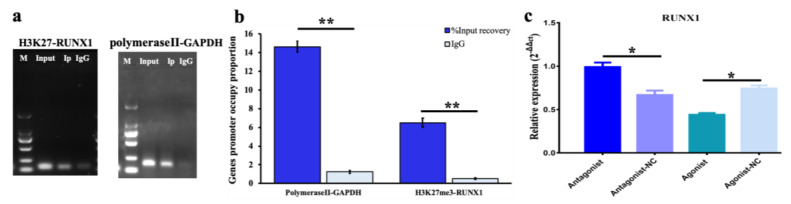
The ChIP-PCR of H3K27me3 in pGCs. (**a**) The occupation of RNA polymeraseII; and H3K27me3 in GAPDH and the RUNX1 promoter by was quantified by assessing the ChIP-PCR products. M denotes DNA marker 2000. (**b**) The plot of the promoter occupancy of a proportion of GAPDH and RUNX1 in the percent of Input recovery and IgG in each ChIP-PCR with the 2^−ΔΔct^ method. The * indicates *p* < 0.05 and ** indicates *p* < 0.01. (**c**) qPCR assay showing the regulation of RUNX1 by H3K27me3.

**Figure 5 genes-11-00495-f005:**
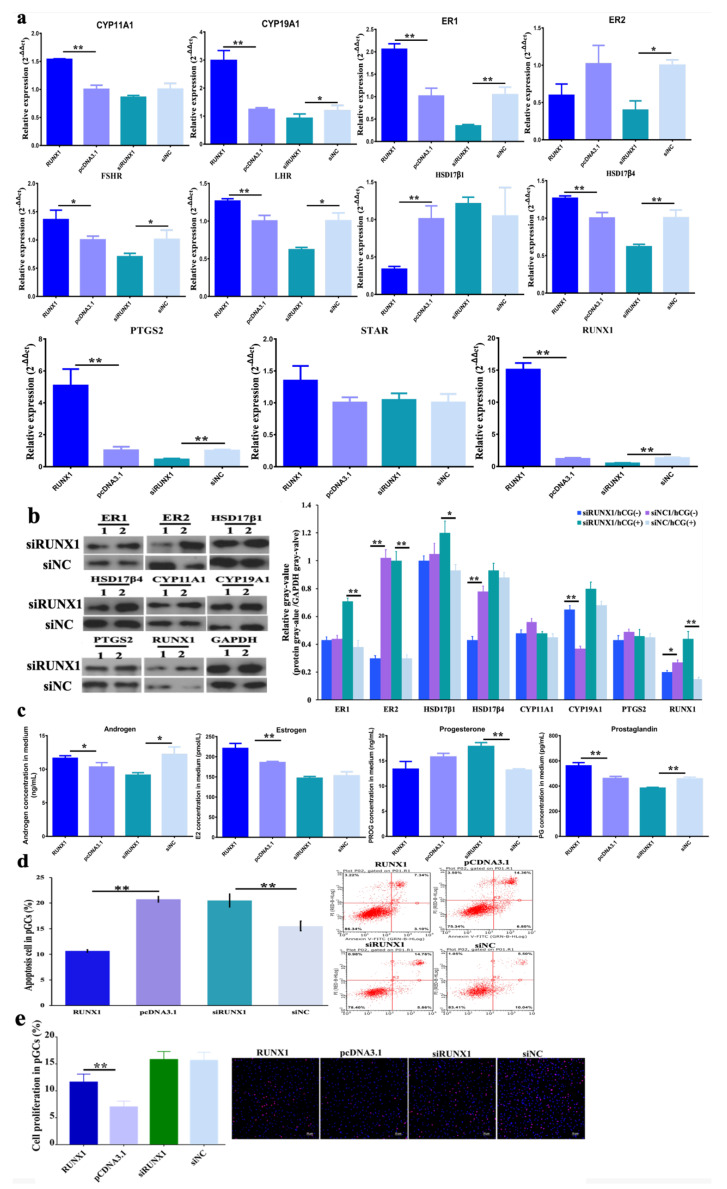
The influence of *RUNX1* on steroidogenesis, cell apoptosis and proliferation in pGCs. (**a**) The expression pattern of steroidogenesis related genes when *RUNX1* was overexpressed or knocked down. The * indicates *p* < 0.05 and ** indicates *p* < 0.01. (**b**) Protein translation of RUNX1, PTGS2, ER1, ER2, HSD17β1, HSD17β4, CYP11A1, and CYP19A1 when pGCs were with/without human chorionic gonadotropin (hCG) in the situation of siRUNX1 or siNC. The numbers 1 and 2 indicate that pGCs were treated without/with hCG. (**c**) The hormone synthesis of androgen, estrogen, prostaglandin, and progesterone in the situation of upregulation and knockdown of *RUNX1* in pGCs. (**d**) The effect of RUNX1 on pGC apoptosis. The pGCs apoptotic proportion in that column plot is a visualization from the sum of the upper and lower right quadrants in the scatter plot. (**e**) The effect of RUNX1 on pGC proliferation. The proportion of proliferated cells in the column equal to the ratio of the numbers of proliferated cells to total cells.

**Figure 6 genes-11-00495-f006:**
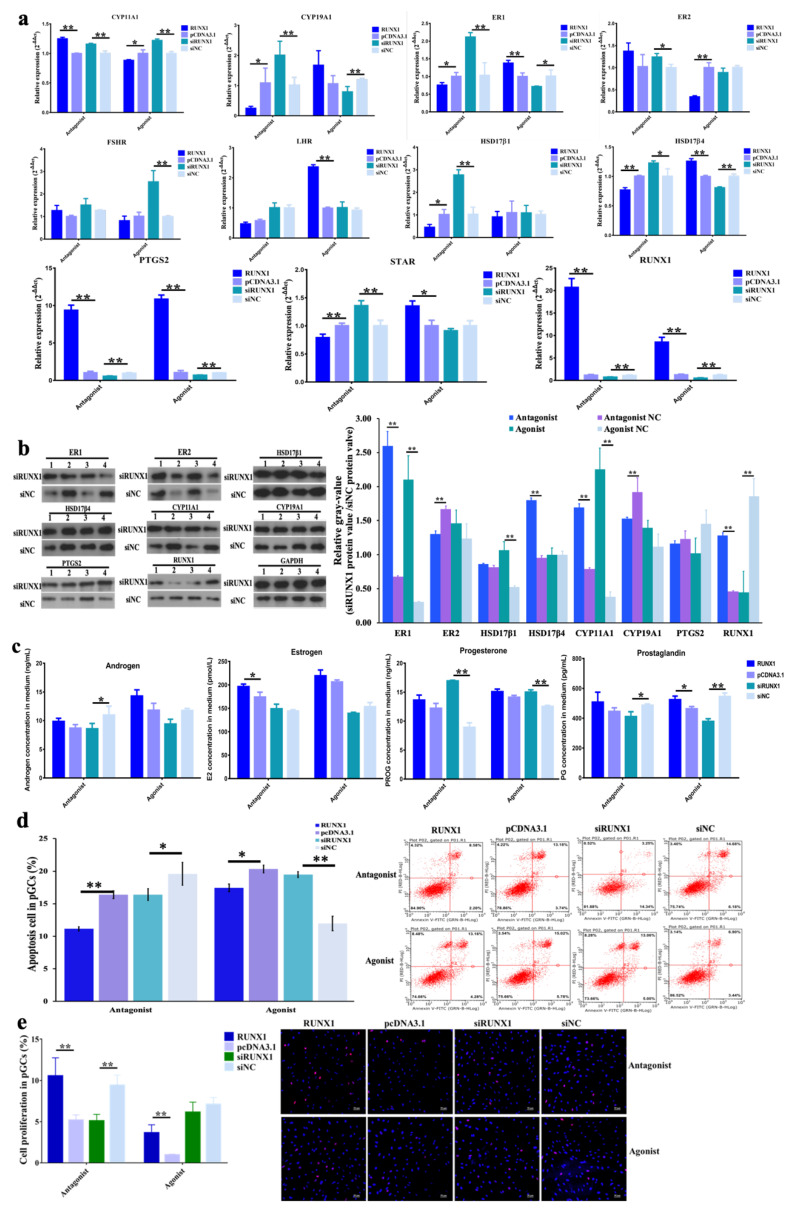
The influence of the H3K27me3-RUNX1 signal on steroidogenesis, cell apoptosis and proliferation in pGCs. (**a**) The expression patterns of steroidogenesis-related genes when RUNX1 was over-expressed and knocked down, and also when treated with antagonist GSK-126 or agonist GSK-J4 at the same time. The * indicates *p* < 0.05 and ** indicates *p* < 0.01. (**b**) Protein translation of RUNX1, PTGS2, ER1, ER2, HSD17β1, HSD17β4, CYP11A1, and CYP19A1. The numbers 1, 2, 3, and 4 indicate treatments with antagonist GSK-126, antagonist-NC, agonist GSK-J4, and agonist-NC. (**c**) The concentrations of androgen, estrogen, prostaglandin, and progesterone in the situation of interfering with the H3K27-RUNX1 signal in pGCs. (**d**) The effect of RUNX1 along with antagonist GSK-126 or agonist GSK-J4 on pGC apoptosis. The pGCs apoptotic proportion in the column plot is a visualization from the sum of the upper and lower right quadrants in the scatter plot. (**e**) The effect of RUNX1 along with antagonist GSK-126 or agonist GSK-J4 on pGCs proliferation. The proportion of proliferated cells in the column is equal to the ratio of total proliferated cells/ total cells.

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
