# Peer review of "Activation of Steroidogenesis, Anti-Apoptotic Activity, and Proliferation in Porcine Granulosa Cells by *RUNX1* Is Negatively Regulated by H3K27me3 Transcriptional Repression"

_genes, 2020, doi:10.3390/genes11050495_

Round 1

Reviewer 1 Report

In this manuscript, authors are presenting an important amount of data on the influence of H3K27me3 on RUNX1 expression in porcine granulosa cells and their effects on genes expression important for steroidogenesis. I have several comments to improve the quality of the manuscript:

Major comments:

The abstract needs more clarifications regarding the results presented. The section "The antagonist GSK-126, with the siRUNX1 signal, and the agonist GSK-J4, with the RUNX1 signal, both led to activation of CYP11A, HSD17β4, and STAR. Hormone levels were altered during the different treatments." is confusing and hormones should be clearly defined.

Decreased H3K27me3 stimulated the expression of CYP11a1, whereas H3K27me3 repression of RUNX1 also resulted in activation of CYP11a1. This seems contradictory and need more clarification (in the abstract).

Regarding RUNX1 results in figure 1b, these should be introduced in the results section, also densitometries should be presented for RUNX1 and compared to H3K27me3 to determine if there is an inverse correlation between H3K27me3 protein counts and expression of RUNX1.

According to figure 2a, it is not clear that the agonist GSK-J4 is consistently increasing H3K27me3 signal. This should be demonstrated more convincingly. On page 9, line 261, HSD17B4 should be indicated as being increased following upregulation of H3K27me3 as this result is not significant according to Fig. 3a.

Overall, the results section should be revised as a great number of differences in expressions are being presented as conclusive data, whereas such differences are not significant according to the figures.

The presentation of data in figures 5b and 6b should be revised and supported by densitometry measurements.

Importantly, authors should explain why the antagonist/agonist effects seen in figure 3 are not observed in figure 6? For example, use of antagonist results in increased expressions of Cyp11A1, ER1 and STAR in Fig 3a, whereas antagonist/pCDNA3 has no effect for Cyp11A1, ER1 and StAR compared to agonist/siNC in Fig 6a.

Minor comments:

"pGCs" should be defined in the title.

Page 2, ln 59: The use of the word "substrate" is not appropriate as progesterone is not a substrate of androstenedione but a precursor.

Page 2, ln 62-65: STARD1 is the same protein as STAR and CYP11A1 is the same protein as P450scc. Hence, these sentences should be revised.

For western blotting methodology, antibodies' dilutions should be indicated.

Page 11, line 293: CYP11A2 should be corrected to CYP11A1.

Author Response

Thank you for your comments and suggestion for the manuscript. we have provided a point-by-point response.

Reviewer 2 Report

Comments on the manuscript Activation of Steroidogenesis, Anti-Apoptotic Activity, and  Proliferation in pGCs by RUNX1 Is Negatively Regulated by  H3K27me3 Transcriptional Repression

This manuscript deals with the epigenetic regulation of steroidogenesis in porcine GC´s. The authors aim to determine a specific metabolic regulation pathway mediated by H3K27me3 enzyme. The amount of work performed is quite impressive, providing a lot of results. To the point that I believe that results could (and maybe should) be discussed more deeply to obtain more reliable and interesting.

However, there is a major issue that I believe should be addressed before acceptance, which is the lack of accurate data regarding the experimental design. Even though the authors explain partially it in the result section, mat and met only describe the methodology employes. I strongly suggest an item describing how the experiments were designed and developed, including, replicates, the number of samples employed in each experiment, etc. As an example, L230 describes results of gene expression but no data on how many replicates or follicles per group were evaluated or if the samples were pooled or not. This is a major issue to determine the reliability of the results, despite the values stated.

Additional comments

Intro

L53 – Author maybe can state that the paragraph is focused on pig physiology

Mat and met

L94 – Why the authors keep the ovaries in saline with ice? Normal protocol is to keep it warm before processing.

L97-98 – Please clarify how the cells were obtained.

L107 – Please detail the  selection criteria employed (I believe that a simple cite is not appropriate)

L112 – The same for WB. Please detail the technique.

Results

There is a general lack of rigor in the results regarding the significance of the data presented. Particularly in WB analysis where most of the results are provided as is, with no significance. It should be stated.

L232 How the morphology is significantly altered? Which test the authors perform.

Fig 1-c The authors should explain as I sated the qWB procedure employed.  

Fig2-a has no info regarding treatments

Fig2b relative expression should be taken with caution when the expression of non-treated is low. Is this the case??

Fig 2 legend is confusing to me. Is the data expressed correctly??

Fig3. In this figure (and these results) is when the lack of information about the experimental design is making more noise. As an example, the authors stated that there is a significant difference in HSD17b4 using antagonist when it seems to be a difference of about 20% between treatments (At least I understand that). For those results, the experimental design should be quite strong and robust. Therefore, the inclusion of experimental that is necessary from my point of view.

L290 – Test stated that STAR and HSD17B1 react equally in the experiment but fig5.a shows that they don’t. Please explain. Also, the p-value was stated as not significant

Fig5b results are not so clear as stated in the text from my point of view. The same for figure 6.b

Fig 5d results are not easy to understand (the scatter plot part). Please explain or remove The same for figure 6.d

L308 – All the paragraph is full of data difficult to follow. Maybe a scheme as a supplementary figure at least??

L325 – Authors cannot state that there is a promotion of 4 molecules by the treatment but only was significant. If statistics say that there are no, it should be taken into account.

Discussion

Part of the discussion is merely a repetition of the results. (for example the first (L356) and third (371) paragraphs. On the contrary, the fourth paragraph (L393) is interesting, very well based on previous studies and clearly stated.  I suggest to the authors to try to improve those weak paragraphs to make a more homogenous and interesting discussion to highlight in a better way all the interesting data produced.

L382 – If the authors stated that “few researchers”, it should be nice that they were cited.

L383 If the phrase start with this report, I don’t understand the cite in the last part of it (20)

L426 The First phrase is not clear to me. Please try to be clearer.

Finally, I´m not a native speaker, and therefore, I´m not qualified to evaluate properly the English quality of the manuscript.

Author Response

(The authors gave the same response as above.)

Reviewer 3 Report

Interesting article, well founded, good identification of the problem and a sound hypothesis to be tested, well declared aims. In general the paper, which is a bit long, is found fully acceptable, provided the authors try to get some English native speaker to go through the text.

Author Response

(The authors gave the same response as above.)

Round 2

Reviewer 1 Report

According to previous comments, the authors greatly improved the content of the manuscript. However, the manuscript should be revised for major text editing. This should be performed by a native english speaking expert.

Author Response

Dear reviewer,

Thanks for your suggestion. The revised report please see the attachment. 
